# Corticosterone Metabolite Concentration Is Not Related to Problem Solving in the Fawn-Footed Mosaic-Tailed Rat *Melomys Cervinipes*

**DOI:** 10.3390/ani12010082

**Published:** 2021-12-30

**Authors:** Misha K. Rowell, Rachel M. Santymire, Tasmin L. Rymer

**Affiliations:** 1College of Science and Engineering, James Cook University, P.O. Box 6811, Cairns, QLD 4870, Australia; tasmin.rymer@jcu.edu.au; 2Centre for Tropical Environmental and Sustainability Sciences, James Cook University, P.O. Box 6811, Cairns, QLD 4870, Australia; 3Biology Department, Georgia State University, Atlanta, GA 30103, USA; rsantymire@gsu.edu

**Keywords:** adrenocortical activity, glucocorticoids, innovation, physiology, rodent, task complexity

## Abstract

**Simple Summary:**

When environments rapidly change, animals must respond through adjustments in their behaviour and cognition, which are largely controlled by physiological processes. In particular, glucocorticoids, such as corticosterone, are important adrenocortical hormones facilitating behavioural and cognitive adjustments. However, we know relatively little about how adrenocortical activity affects problem solving in animals. We therefore studied whether adrenocortical activity was related to problem solving in a native Australian rodent, the fawn-footed mosaic-tailed rat (*Melomys cervinipes*). We collected faecal samples and measured glucocorticoid metabolite hormone concentrations as a measure of adrenocortical activity using an enzyme immunoassay. Corticosterone metabolite concentrations were then compared to problem solving measured using five food-baited puzzles and one escape-motivated puzzle. Interestingly, adrenocortical activity was not related to how quickly problems were solved, or how much the rats interacted with the problems. However, given that adrenocortical activity is involved in multiple processes, future studies should also compare behaviour to other physiological measures.

**Abstract:**

Animals can respond physiologically, such as by adjusting glucocorticoid hormone concentrations, to sudden environmental challenges. These physiological changes can then affect behavioural and cognitive responses. While the relationships between adrenocortical activity and behaviour and cognition are well documented, results are equivocal, suggesting species-specific responses. We investigated whether adrenocortical activity, measured using corticosterone metabolite concentration, was related to problem solving in an Australian rodent, the fawn-footed mosaic-tailed rat (*Melomys cervinipes*). Mosaic-tailed rats live in complex environments that are prone to disturbance, suggesting a potential need to solve novel problems, and have been found to show relationships between physiology and other behaviours. We measured problem solving using five food-baited puzzles (matchbox and cylinder in the home cage, and activity board with pillars to push, tiles to slide and levers to lift in an open field), and an escape-motivated obstruction task in a light/dark box. Faecal samples were collected from individuals during routine cage cleaning. Adrenocortical activity was evaluated non-invasively by measuring faecal corticosterone metabolites using an enzyme immunoassay, which was biochemically and biologically validated. Despite varying over time, adrenocortical activity was not significantly related to problem solving success or time spent interacting for any task. However, as adrenocortical activity is reflective of multiple physiological processes, including stress and metabolism, future studies should consider how other measures of physiology are also linked to problem solving.

## 1. Introduction

Throughout an animal’s life, it will experience changes in its immediate environment. As it grows and develops, a cascade of physiological changes will influence how it responds to its environment [1]. Likewise, when the animal disperses from its natal territory, it will experience a suite of novel challenges, including increased competition and predation risk. To cope with these challenges, animals can rapidly adjust their behavioural (e.g., exploration) and cognitive (e.g., learning) responses, which are both underpinned by the animal’s adrenocortical activity [2].

Adrenocortical activity is the change in an animal’s glucocorticoid (e.g., cortisol and corticosterone) concentrations [3]. These physiological changes could be a marker for an animal’s metabolism, which is an animal’s energetic state [1]. Adrenocortical activity can therefore influence how much energy and/or time an animal can invest in simply growing and maintaining its own condition [4], foraging, defending territory, migrating or mating [1]. Adrenocortical activity could also indicate an animal’s stress levels, as glucocorticoids are released after various stressors [5]. Consequently, an individual’s adrenocortical activity can impact its behavioural [6] and cognitive outputs [7]. For example, glucocorticoids mobilise glucose [8], providing energy for appropriate behavioural (e.g., increased foraging; [9]) and cognitive (e.g., rapid learning; [10]) responses to threats [11]. However, the level of individual variation in adrenocortical activity is not well known for many species. This variation could contribute to differences in how individuals cope with environmental challenges, such as the ability to solve problems [12].

Simply defined, problem solving is an animal’s ability to overcome an obstacle to obtain a reward [13]. Problem solving can be innovative, where an animal uses a new behaviour, or an existing behaviour in a new context, to solve the problem [14], or can rely on forms of learning, such as trial-and-error [15] or imitating other individuals [16]. Individual variation in problem solving is well documented in many species, often due to underlying individual differences in motor skills, personality and/or cognitive ability [17]. However, physiological responses, such as adrenocortical activity, could also contribute to individual differences in problem solving [18]. For example, problem solving success in a foraging task presented to pheasant chicks (*Phasianus colchicus*) was affected by differences in motivation [19], which is regulated by the neurotransmitter dopamine [20]. However, the direct links between physiology and problem solving performance are poorly studied. 

Currently, studies relating adrenocortical activity to problem solving ability show mixed results. For example, Bókony et al. [12] found that house sparrows (*Passer domesticus*) with lower corticosterone concentrations solved complex problems faster than birds with higher corticosterone concentrations. However, horses (*Equus callabus*) that were capable of innovating had significantly higher corticosterone concentrations than horses that did not innovate [21], and blood glucose and ketone concentrations were not related to problem solving in African striped mice (*Rhabdomys pumilio*) [22]. This suggests that the relationship between adrenocortical activity and problem solving may be species-specific, and that the methods of measuring this response could impact the results.

Therefore, we investigated whether adrenocortical activity, assessed using corticosterone metabolite concentration, was related to problem solving performance in a native Australian rodent, the fawn-footed mosaic-tailed rat (*Melomys cervinipes*). Mosaic-tailed rats are medium-sized (72.9 ± 12 g) murid rodents found in forests along the eastern coast of Australia [23]. These habitats are complex, showing spatial and temporal variation in food availability, indicating that mosaic-tailed rats likely experience variations in metabolic demand. Furthermore, these habitats often experience high levels of disturbance, yet mosaic-tailed rats also thrive in these conditions, indicating a good capacity for problem solving. In support of this, mosaic-tailed rats are capable innovators [24]. Previous studies have found that more exploratory mosaic-tailed rats are better problem solvers than avoidant individuals [25]. Mosaic-tailed rat personality has also been linked to some physiological measures (glucose and testosterone concentrations, [26]), suggesting a potential for adrenocortical effects on problem solving. We measured problem solving performance using six problem types of different complexities in different contexts, and we collected faecal samples from each individual outside of these contexts to assess corticosterone metabolite concentrations, and therefore, adrenocortical activity levels. As adrenocortical activity could reflect variations in metabolism [10] or stress [11], both of which are known to impact behaviour [7,27] and cognition [28], we predicted that individual variation in adrenocortical activity would reflect individual variation in problem solving ability, but due to mixed results (e.g., [21,22]), we made no a priori predictions about the direction of this relationship. 

## 2. Materials and Methods

### 2.1. Ethical Note

The research complied with the ABS/ASAB guidelines for the ethical treatment of animals [29] and the Australian Code for the Care and Use of Animals for Scientific Purposes [30]. The study was approved by the Animal Ethics Screening Committee of James Cook University (clearance number: A2539). We observed all animals daily, and animals were weighed every two weeks to monitor condition. No animals exhibited signs of extreme stress (e.g., prolonged vocalisation, seizures, sudden weight loss) during testing or after testing. Individuals were returned to their home cages following testing and were monitored for any abnormal behaviours (of which none were observed).

### 2.2. Husbandry

Animals used in this study were 25 wild-caught individuals from a free-living population (14 male and 11 female) and 29 F1 or F2 captive-born (15 male and 14 female) individuals from the mosaic-tailed rat breeding colony at James Cook University, Cairns, Australia. F1 individuals were individuals born in captivity to 11 wild-caught females, while F2 individuals were born to F1 females in captivity. This represents a larger sample size (*n* = 54) than some other studies examining the relationship between problem solving and glucocorticoid concentrations (e.g., 16 individuals in Esch et al. [21]; 23 individuals in Pfeffer et al. [31]). All individuals were tested as adults (i.e., sexually mature; range: 7 months to >2 years of age at first test). Mosaic-tailed rats were kept under partially controlled environmental conditions (natural lighting from a large window; 22–26 °C; 50–65% relative humidity) in a laboratory for at least 7 months before being tested.

As mosaic-tailed rats are solitary [32], they were individually housed in wire frame cages (34.5 cm × 28 cm × 38 cm) with deep plastic bases (34.5 cm × 28 cm × 11 cm) containing ± 10 cm of wood shavings. Individuals were each given a cylindrical plastic nest box (11 cm × 11 cm × 20 cm), a handful of pasture hay and two pieces of paper towel for nesting material. A cardboard roll was provided for enrichment. As mosaic-tailed rates are semi-arboreal [32], cages were equipped to enable climbing, with a wire platform near the top of the cage and sticks from the base to the top of the cage. Approximately 5 g of mixed seed and rodent chow (Rodent Origins, Vetafarm, Wagga Wagga, Australia) and 5 g of vegetable/fruit (e.g., sweet potato, apple) were given to each individual daily. Average animal mass during the testing period was 91.05 ± 1.28 g. On days of problem solving testing, food was only given to the animals once testing was completed to motivate animals to participate in the test. All individuals were therefore deprived of food for approximately 24 h prior to testing. Water was available ad libitum.

### 2.3. Problem Solving Tests

Tests were conducted between August 2018 and August 2020. Individuals were tested in a random order, and received the different problem tasks in a random order, except for the Trixie dog activity board, which was presented last due to its complexity and the need to habituate animals first to the novel arena in which it was presented (see below). Individuals were rested for at least 24 h between each test. Testing occurred during the peak period of mosaic-tailed rat activity (between 6:00 p.m. and 10:00 p.m.; [33]) under red light (except for the obstruction test), which does not affect mosaic-tailed rat behaviour [34]. Tests were recorded with a Sony HDR-CX405 Camcorder from above. Behavioural data were then extracted from videos using the video analysis software BORIS (version 7.9.6) [35]. Mosaic-tailed rats were tested individually and returned to their home cage and fed immediately after testing.

To gain a comprehensive measure of the problem solving abilities of individuals, we assessed problem solving using five foraging-motivated problem-solving tests and one escape-motivated problem-solving test. These problems differed in complexity across three contexts (home cage, open field arena, or light/dark box). Animals were not trained in any test, and were only presented with each test once, as we were interested in the natural problem solving abilities of individuals. The methods are fully outlined in Rowell and Rymer [24]. If testing occurred outside of the home cage (i.e., the Trixie Dog Activity Board Level 2 and the obstruction task), mosaic-tailed rats were removed from their cage by gently guiding them into a plastic cup using a piece of cardboard. The individual was then released into the testing arena on the side opposite the problem. The individual was returned to its home cage after testing in the same manner. For all tests, we measured the latency to solve the puzzle once the individual began interacting with it, the time spent interacting with the puzzle, and whether the puzzle was successfully solved.

#### 2.3.1. Problem Solving in the Home Cage

Briefly, we presented mosaic-tailed rats with two food-baited puzzle boxes in their home cages (presented on separate nights). The matchbox task consisted of a cardboard matchbox (solved by pushing or pulling out the sleeve, or chewing through the box) and the cylinder task was a plastic cylinder (solved by pulling or pushing through the open end that was covered with tinfoil). An amount of 2 g of fruit (e.g., banana) was used as a reward in these tests.

#### 2.3.2. Problem Solving in an Open Field

We also presented mosaic-tailed rats with three simultaneous problems on a food-baited Trixie Dog Activity Board Level 2 (two pillars to push, two tiles to slide and two levers to lift) placed in an open field arena. Two sunflower seeds were placed under each task on this board, as fruit could have become lodged in the mechanisms. Although presented at the same time, as they were located on the same board, these tasks required different methodologies to solve and differed in complexity. Mosaic-tailed rats were given 30 min to interact with and solve the puzzles (they only had to solve one of each type (e.g., one pillar) to be classified as a successful solver of that task).

#### 2.3.3. Problem Solving in the Light/Dark Box

We also used an obstruction task to measure escape-motivated problem solving in a light/dark box. Mosaic-tailed rats had to either push or pull a crumpled piece of paper out of a doorway to escape to the dark compartment. This was presented to the mosaic-tailed rats for three, 3-min rounds in a single testing session to increase the chance of participation, as this was a stressful test because of a bright LED light trained on the light compartment of the box [24]. In terms of complexity, the pillar task was the easiest to solve, followed by the cylinder, matchbox, obstruction, tile, and lever tasks [24].

### 2.4. Faecal Sample Collection and Extraction

Faecal samples were collected from individuals between September 2019 and June 2020 during routine cage cleaning. Some individuals (*n* = 42) had two sets of samples collected and analysed (one from 2019 and one from 2020). Faecal samples were only collected once individuals had experienced all problem solving tests. Mosaic-tailed rats were placed in a plastic tub while the cage was cleaned (maximum 20 min process) and any faecal boli excreted during this time (that were uncontaminated by urine) were collected. Cages were cleaned between 9:00 a.m. and 11:00 a.m. once per fortnight. At least two sessions of cleaning were required to collect sufficient faecal boli for each individual (we required at least 1 g of faeces per individual). Although the gut retention time of mosaic-tailed rats is unknown, collection of faeces in the morning likely corresponded with their main activity period (night time). Faecal samples were stored in a plastic Eppendorf tube at −20 °C. We used several faecal samples collected during multiple routine cage cleaning episodes as these would likely provide a better representative of general adrenocortical activity, rather than a stress response to the problem solving tests. Due to the gut passage time, sampling during cage cleaning was not expected to significantly impact hormone concentrations.

To biologically validate faecal corticosterone metabolite measurements in this species, we collected faecal samples from some of the individuals (*n* = 11) while they were undergoing repeated sessions of behavioural and cognitive testing for another study that will be published separately. These tests were conducted outside of the home cage exposing individuals to novel objects in a novel arena and were thus considered to be stressful for the individuals. Testing occurred at the beginning of the dark cycle (i.e., between 6:00 p.m. and 7:00 p.m.). Individuals received multiple tests approximately 24 h apart and had at least two faecal samples collected. Fresh faecal samples were collected immediately following the testing sessions. These samples were collected for another study that will be published separately. 

Faecal samples were prepared according to Murray et al. [36]. Briefly, samples were flash-frozen for 15 min at −80 °C, and then dried overnight in a ScanSpeed 40 speed vacuum concentrator spun at 1000 rpm and connected to a Scanvac CoolSafe condenser at ~100 °C. Samples were then weighed to the nearest 0.01 mg. An 80% ethanol (100% ethanol: distilled water) solution was added to each sample so that the sample was at a 1:10 concentration. Faecal boli were physically broken down into small pieces by using tweezers to push the boli against the sample tube walls. Tweezers were thoroughly wiped with ethanol in between each sample to avoid cross-contamination. Samples were vortexed and placed on a rotator overnight. Samples were then centrifuged at 15,000 rpm for 15 min. We created a dilution by pipetting 2 μL of the 80% ethanol solution from each sample into 998 μL of hormone kit assay buffer with a final sample dilution of 1:500. All samples were then vortexed again.

### 2.5. Faecal Corticosterone Metabolite Quantification

We used corticosterone enzyme-linked immunosorbent assay (ELISA) kits (ADI-900-097, Enzo Life Sciences, Farmingdale, New York, NY, USA), following the manufacturer’s instructions, to analyse faecal corticosterone metabolites in each sample. Samples were plated in duplicate to measure intra-assay variation (average 11.4%). A single sample was repeated across plates to measure inter-assay variation (2.2% coefficient of variation). The cross-reactivities of corticosterone antibodies are reported in detail in previous studies [37]. Sample concentrations were multiplied by 500 to account for the dilution.

As per the ELISA kit instructions, plates were read on a POLARstar Omega (BMG Labtech, Ortenberg, Germany) plate reader to measure the optical density of each well. We used Omega software v5.11 (BMG Labtech) and MARS Data Analysis software v3.20 R2 (BMG Labtech) to compare the optical densities to the standard curve. One individual was excluded from all statistical analyses (see below) due to an extremely high (173 × more than the group average) corticosterone metabolite concentration, suggesting an error during the plating process. Despite the high corticosterone concentration, the problem solving abilities of this individual were similar to other animals included in the study (solved 3 of the 6 tasks, average 130 s latency to solve 3 tasks).

### 2.6. Statistics

Statistical analyses were performed using RStudio (version 1.0.153; https://www.rproject.org; R version 4.0.2, https://cran.rstudio.com, 22 June 2020). Data are available in the Appendix A. The model-level significance was set at α = 0.05. Data and model residuals were tested for normality (Shapiro–Wilk test). The descdist function (‘fitdistrplus’ package, [38]) was used to find the distribution of best fit for each response variable used in the regression models. In all models, animal birth origin (captive born vs. wild caught) and sex were included as fixed factors. Body mass and 2020 faecal corticosterone metabolite concentrations (non-transformed, unless otherwise stated) were included as continuous predictors. Linear models were checked for homoscedasticity of residuals using the gqtest function (‘lmtest’ package, [39]) and for a linear relationship with continuous predictors. The emmeans function (‘emmeans’ package, [40]) was used to calculate the means and standard errors of variables included as dependent variables in the models.

We used a paired *t*-test to test the biological validity of faecal metabolite measurement in this species. We compared the two measures of non-transformed faecal corticosterone metabolite concentrations between the samples collected after undergoing a stressor and samples collected during cage cleaning from the same individuals.

We then considered the intra-individual variation in corticosterone metabolite concentration using a test of repeatability (‘rptR’ package, [41]) to analyse whether the corticosterone metabolite concentration of individuals (*n* = 42) significantly changed from 2019 to 2020. We also considered inter-individual variation of the 2020 samples, as this was the larger sample size (*n* = 53). We used a linear model with 2020 corticosterone metabolite concentration (log-transformed) as the dependent variable, and individual mass, sex, and birth origin as independent variables.

We ran separate Cox’s proportional hazards models for each year of physiological sampling to analyse the relationship between problem solving latency and the corticosterone metabolite concentrations (2019 and 2020) across all six solving tasks (‘survival’ package [42]; ‘survminer’ package, [43]). We report the hazard ratios (differences between groups in the limit of the number of events per time/number at risk; *e^b^*) with confidence intervals and *p* values for these models (as per [12]). For the puzzles presented in the open field arena (the pillar, tile and lever tasks on the Trixie Dog Activity Board), we used the latency to solve the first of each puzzle type, as puzzles were given in duplicate (e.g., whichever tile was slid open first). The maximum value was given if an individual did not solve a task (i.e., 1800 s; [44]) or did not participate in the test (i.e., did not interact with the task; cylinder = 1 individual, matchbox = 2 individuals, obstruction = 11 individuals). The maximum latencies were treated as censored data in the Cox’s proportional hazards models.

The data for time spent interacting were not normally distributed. Therefore, the 2020 data were transformed using the orderNorm function (‘bestNormalize’ package, [45]). The data were then analysed using a linear model with a Gaussian distribution (‘lme4′ package, [46]). The 2020 corticosterone metabolite concentration was included as a continuous predictor, and problem type was included as a fixed factor. The 2019 data could not be transformed, and so a separate beta regression model (‘betareg’ package, [47]) was conducted with the proportional time interacting, with the 2019 corticosterone metabolite concentration included as a continuous predictor, and problem type as a fixed factor. If two replicas of the same puzzle were presented to the mosaic-tailed rats (e.g., pillar, tile, lever and obstruction tasks), we used the average time spent interacting between the replicates.

As per the descdist function, we used a general linear model (‘nlme’ package [48]) with a Poisson distribution to assess the effect of 2020 corticosterone concentration on overall problem solving performance (i.e., the total number of puzzle types solved). The model was tested for overdispersion using the dispersiontest function (‘AER’ package, [49]). Individuals were considered to have solved a puzzle if at least one repeat (e.g., one of two tiles) was solved. There were six types of puzzles that could be solved (matchbox, cylinder, obstruction, pillar, tile, lever).

## 3. Results

### 3.1. Biological Validation of the Assay to Measure Faecal Corticosterone Metabolites

Behavioural and cognitive testing resulted in a significant increase in adrenocortical activity in mosaic-tailed rats (t_10_ = 4.26, *p* = 0.002, stressed = 137.47 ± 28.12 ng/g faeces, control = 64.81 ± 13.63 ng/g faeces), with faecal corticosterone metabolite concentration being 2.1 × higher when individuals were undergoing testing than when they were not (Figure 1).

### 3.2. Individual Variation in Corticosterone Metabolite Concentration

Individual corticosterone metabolite concentrations (measured from faecal samples collected during cage cleaning) were not repeatable between 2019 and 2020 (R = 0.09, *p* = 0.281). The average faecal corticosterone metabolite concentration was 138.30 ± 21.82 ng/g faeces for mosaic-tailed rats sampled in 2019, and 79.86 ± 12.26 ng/g faeces for mosaic-tailed rats sampled in 2020. Individual concentrations ranged between 11.58 and 494.25 ng/g faeces. Faecal corticosterone metabolite concentrations were similar between individuals of different sexes (F_1,53_ = 0.05, *p* = 0.826; male = 82.31 ± 17.69 ng/g faeces, females = 67.52 ± 9.50 ng/g faeces), birth origins (F_1,53_ = 0.06, *p* = 0.813; captive born = 79.94 ± 8.20 ng/g faeces, wild caught = 70.17 ± 10.59 ng/g faeces), and mass (F_1,53_ = 0.34, *p* = 0.562).

### 3.3. Latency to Solve Problems

Regardless of the year of sampling, the latency to solve the six problems was not significantly related to corticosterone metabolite concentration (2019: *e^b^* = 0.99 [0.99, 1.00], *p* = 0.380; 2020: *e^b^* = 0.99 [0.99, 1.00], *p* = 0.397), sex (2019: *e^b^* = 0.92 [0.64, 1.84], *p* = 0.929; 2020: *e^b^* = 1.03 [0.59, 1.77], *p* = 0.795), birth origin (2019: *e^b^* = 1.10 [0.61, 1.96], *p* = 0.758; 2020: *e^b^* = 0.95 [0.61, 1.49], *p* = 0.822), mass (2019: *e^b^* = 0.99 [0.95, 1.03], *p* = 0.584; 2020: *e^b^* = 1.00 [0.97, 1.03], *p* = 0.991) or problem type (2019: *e^b^* = 1.12 [0.15, 8.85], *p* = 0.912; 2020: *e^b^* = 0.98 [0.30, 3.22], *p* = 0.977).

### 3.4. Time Spent Interacting with Problems

Time spent interacting with the tasks was significantly related to the problem type (2019: χ^2^_5_ = 65.29, *p* < 0.001; 2020: *F*_1,5_ = 40.01, *p* < 0.001), with individuals interacting significantly less with the pillar and tile tasks than all other tasks (Figure 2). Corticosterone metabolite concentration (2019: χ^2^_1_ = 1.81, *p* = 0.178; 2020: *F*_1,318_= 0.50, *p* = 0.482), birth origin (2019: χ^2^_1_ = 3.36, *p* = 0.067; 2020: *F*_1,318_ < 0.01, *p* = 0.950), mass (2019: χ^2^_1_ = 0.09, *p* = 0.762; 2020: *F*_1,318_ = 3.89, *p* = 0.050), and sex (2019: χ^2^_1_ = 2.62, *p* = 0.106; 2020: *F*_1,318_ = 3.65, *p* = 0.057) were not significantly related to how much time the mosaic-tailed rats spent interacting with the problems.

### 3.5. Overall Solving Performance

The total number of problems solved was not significantly related to corticosterone metabolite concentration (χ^2^_1_ = 0.43, *p* = 0.512), sex (male = 1.42 ± 0.11, female = 1.33 ± 0.10; χ^2^_1_ = 0.13, *p* = 0.720), birth origin (captive born = 1.41 ± 0.10, wild caught = 1.34 ± 0.11; χ^2^_1_ = 0.65, *p* = 0.420) or mass (χ^2^_1_ = 0.04, *p* = 0.837).

## 4. Discussion

In this study, we explored the relationships between adrenocortical activity (corticosterone metabolite concentration) and problem solving in a native Australian rodent. We found that, individuals varied in their adrenocortical activity over time, possibly due to seasonal variation (e.g., house sparrows [50]), age effects (e.g., chickens (*Gallus gallus domesticus*), [51]), variation in sex hormones (e.g., Wistar rats (*Rattus norvegicus*), [52]), or unintended methodological variation. Despite these intra-individual differences, there was no significant variation in adrenocortical activity between individuals of different birth origin, sex, or mass. While we found a difference in metabolite concentrations between stressed and control animals, and we did not find an effect of yearly corticosterone variation on problem solving, we must express some caution. The collection time for the stressed and control samples differed due to behavioural testing being conducted at the beginning of the dark phase and cages being cleaned (and control samples collected) at the start of the light phase. A circadian rhythm influenced faecal corticosterone metabolite levels in laboratory rats, and metabolite levels peaked towards the end of the dark phase, and were lowest at the beginning of the light phase [53]. Therefore, we would expect methodological differences to reflect either (1) no variation between samples, as the interaction between stressful condition and circadian rhythm would “cancel” each other out or (2) that the samples collected during routine cage cleaning would show higher concentrations, reflecting the peak circadian rhythm activity period (the night before). However, our results indicate the opposite pattern (i.e., higher concentrations collected during the testing phase), indicating that the difference in metabolite levels between stressed and control groups most likely reflects the response to the testing stressor. While blood samples collected from this species during the dark phase show a similar increase in corticosterone concentrations due to stress [26], we recommend that future studies should collect faecal samples during all phases. For both 2019 and 2020 sampling points, faecal sample collection periods varied between individuals, as the number of faecal boli individuals produced varied, with some individuals producing more in each sampling session than others. Variation in adrenocortical activity between samples could therefore also reflect seasonal variation [50] or differences in other physiological cycles (e.g., oestrous cycle, [52]). Future studies should consider adjusting the sampling period for all individuals to minimize these possible influences.

Contrary to expectations, adrenocortical activity was not related to the latency to solve problems or the total number of problems solved. For the simple problems, these results are consistent with Bókony et al. [12], who also found corticosterone concentration was not related to the latency to solve simple tasks in house sparrows. This could suggest that these tasks may not solely rely on cognitive processes (e.g., trial-and-error learning) to solve, which are known to be impacted by adrenocortical activity (e.g., metabolism [27]). Instead, the successful solving of simple or moderate tasks may have relied on individuals being motivated and persistent [54], which is related to other physiological processes (e.g., dopamine release, [55]).

However, our results for the more complex problems were not supported by previous work. Other studies have found that adrenocortical activity was related to problem solving performance in complex problems, with higher corticosterone concentrations increasing problem solving performance in some species (e.g., greylag geese (*Anser anser*), [31]; horses, [21]) and decreasing performance in others (e.g., house sparrows, [12]). However, problem solving in African striped mice was not related to physiological measures, including glucose and ketone concentrations as a measure of metabolism [22]. The lack of relationship between adrenocortical activity and complex problem solving performance in mosaic-tailed rats could indicate that, as for simple tasks, problem solving might be more reliant on other physiological processes.

This lack of consistency in results could also be due to the complex effects that corticosterone may have on an animal’s physiology and behaviour, as well as a misunderstanding of what these measures represent [56]. While higher corticosterone concentration can indicate heightened metabolic response [56], it may also be indicative of heightened stress [57] or eustress (i.e., stimulation or arousal, [58,59]). As such, using only corticosterone concentration as a physiological measure may not be adequate to fully determine an animal’s adrenocortical activity level (e.g., metabolic condition or stress level) [56]. Future studies should, therefore, consider measuring other physiological measures (e.g., oxidative stress and parasite load as indicators of a negative state, [12]; salivary proteins as indicators of a positive state, [60]), and how these relate to problem solving performance.

Similar to what has been found in other studies [24], mosaic-tailed rats differed in the amount of time spent interacting with different types of problems. Individuals spent significantly less time interacting with two of the problems in the open field (the pillar and tile tasks) than in the home cage (matchbox and cylinder) and light/dark box (obstruction) problems. The difference in interacting behaviour is likely a function of test context and complexity. Individuals may have been more active and less stressed in their home cages [61], leading to higher times interacting with these problems. In contrast, individuals may have been more motivated to solve and escape the stressful conditions of the obstruction task [62] than in the open field, also leading to higher times spent interacting with these problems. Within the open field, the lever task was the most complex task, and individuals needed to interact with it more to solve it [24], resulting in the time interacting being similar between this task and the tasks in the other contexts.

Overall, we found that adrenocortical activity, as measured using corticosterone metabolite concentrations, was not significantly related to the latency to solve problems or time spent interacting with problems by fawn-footed mosaic-tailed rats. This could indicate that adrenocortical activity, possibly indicating stress, eustress, or metabolism, does not interfere with the capacity to problem solve in mosaic-tailed rats. Due to the complex nature of adrenocortical processes, future studies should consider including other markers of physiology (e.g., oxidative stress or positive state) when comparing physiology to behaviour and cognition in this species.

## Figures and Tables

**Figure 1 animals-12-00082-f001:**
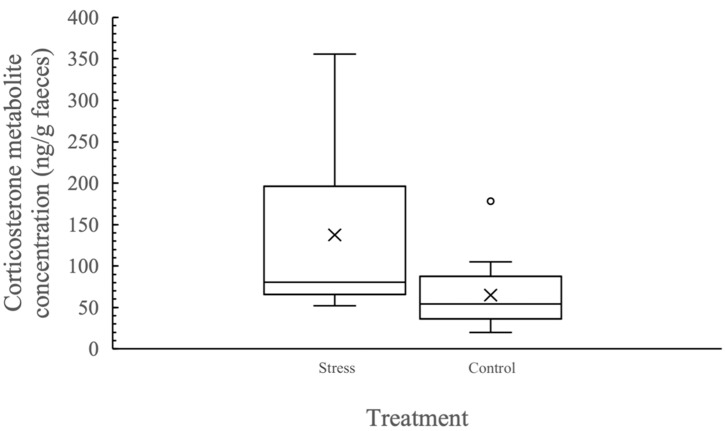
Boxplot graphs of faecal corticosterone metabolite concentrations (ng/g faeces) of individual fawn-footed mosaic-tailed rats (*Melomys cervinipes*) when undergoing testing (Stressed) and when not being tested (Control). The mean for each group is indicated by X.

**Figure 2 animals-12-00082-f002:**
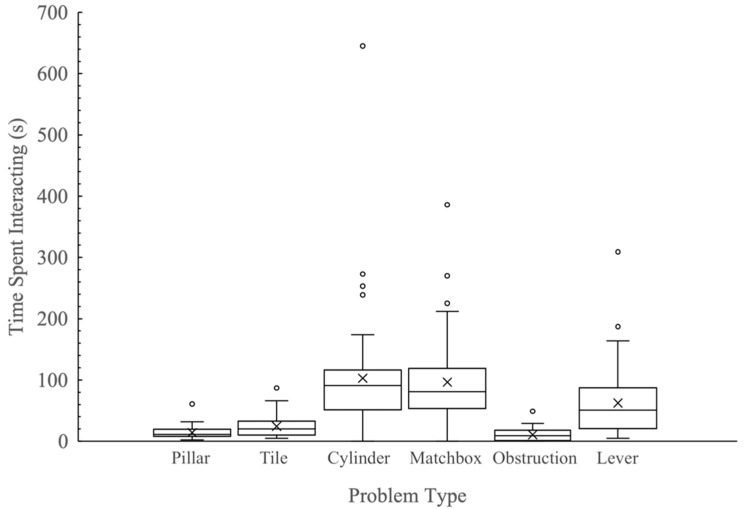
Boxplot graphs of the relationship between problem type and time spent interacting (s) with the tasks of fawn-footed mosaic-tailed rats (*Melomys cervinipes*). The mean for each group is indicated by X.

## Data Availability

Please contact the authors for access to the data.

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
