# Peer review of "Corticosterone Metabolite Concentration Is Not Related to Problem Solving in the Fawn-Footed Mosaic-Tailed Rat Melomys Cervinipes"

_animals, 2021, doi:10.3390/ani12010082_

Round 1
Reviewer 1 Report
Generally, I appreciate the great length at which the manuscript has been revised, but two major issues remain: the statistical analyses are inappropriate, and the interpretation of the fecal hormone measurements (and thus their relationship with problem solving) is not clear.
You used negative binomial models to analyze problem-solving latencies, which is problematic for the following reasons. First, the negative binomial distribution assumes count data, i.e. it models the frequency with which an event occurs. Solving latency is not such a variable. Second, as I explained in my first review, treating capped latencies as real values is incorrect. The fact that several earlier papers made the same mistake is not a valid reason for replicating their error. The solving time of individuals that did not solve the task is unknown, so it should not be treated as if it was 1800 sec, and it should not be excluded either because that would cause selective bias. I see no acceptable reason for not using Cox’s proportional hazards models or similar approaches here.
Based on the Referees' criticisms, the manuscript is now re-written to focus on the metabolic effects of glucocorticoids rather on the role they play in the stress response. Although fecal cort has the potential to reflect individual differences in adrenocortical activity, this cannot be taken for granted without validation studies, which seem to be missing in the species studied here. I'm not sure that totally ignoring the stress aspect is the right solution, but the new focus on metabolism is similarly problematic. Instead of equating fecal cort with chronic stress, now you are simplifying it to glucose levels. I disagree with Referee 2 that you measured "baseline" corticosterone. Fecal cort is an integrative measurement that depends not only on baseline levels but also on any stress-induced elevations that happened since the last defecation. Like all integrative measures, it also depends on the length of time period over which the amount is integrated (e.g. time between defecations) and the speed with which cort is deposited per unit time (e.g. gut passage time) and the amount of matrix it is deposited in (e.g. fecal volume and composition). Without validation, it is impossible to know what exactly the measured values mean (not only internal validation of the cort assay itself but also external validation of the biological meaning of the measured values). These issued are explained in several papers, e.g. Touma & Palme (2005) and Gormally & Romero (2020). If you can, please report in the manuscript how long after the last problem-solving test the fecal sampling began. Without this information, it is unclear how much your cort values may have anything to do with potential stress or metabolic demands during the tasks. A further difficulty with interpreting your cort values is that you did not record exactly how many feces-collecting sessions were required for each individual. Because fecal sample collection dates were separated by 2 weeks, some individuals may have been sampled over a much longer period of time than others. Therefore, any temporal change in fecal cort (e.g. due to seasonal change or to any diminishing effect of the testing procedures) could have confounded the measured individual variation in cort. For these reasons, it is unclear if the cort measurements express consistent individual differences in metabolism, differences in the amount of stress experienced, or artificial differences due to sampling design. These issues should be acknowledged in the manuscript, and the interpretation of fecal cort should be more careful. For example, it cannot be concluded that rats with lower/higher fecal cort had "lower metabolic states" (L21-22, L37-38). The phrasing of the text is still plagued with incorrect interpretations, e.g. in L37-38: you did not measure "metabolic condition" during the tests, and even if you did, a correlation between two variables does not necessarily mean that one "impairs" the other.
L63-65: Yes, but the opposite relationship is also possible, i.e. that a lower metabolic state makes it more important for the individual to solve novel challenges, e.g. a hungrier animal might have a higher need for acquiring food. There have been many results both pro and contra to this; see the "necessity or capacity" debate in the animal innovation literature.
L71: Another important, consistent underlying factor is motor variability or diverse manipulative skills (Griffin & Guez 2014).
L80: The cited study did not measure baseline corticosterone concentrations. They measured feather corticosterone concentrations, which is an integrative measure similarly to fecal cort.
L85-86: At least as likely is the explanation that the results vary with the methods (e.g. which physiological variable was measured and how)!
L87, L101 etc.: You did not measure baseline concentrations: see above!
L122-124: I agree with your rebuttal to Referee 3 that your sample size is not problematic. However, please rephrase this sentence, because this is not generally true. For example, Bokony et al. (2014) had n=97 (this paper is cited in the manuscript).
L225-227: This is okay, but still, please provide some information in the manuscript on the problem-solving performance of the excluded individual. How many tasks did it solve? Was it a particularly fast or slow solver?
L248: It should be explicitly stated in the text that non-participating individuals (i.e. those that did not solve not because they were unsuccessful but because they did not interact with the task) were not excluded but were given the maximum latency. Please report how many of the maximal latencies belonged to non-participating individuals.
L256-258: The time spent interacting with the task is not a count variable. Please provide justification in the manuscript that this variable fits the negative binomial distribution. Model diagnostics for such models should be done with the DHARMa package in R.
Results throughout:
- It is obvious from Figure 1 that the latency data should be analyzed with Cox models, and mean±SE values are inappropriate for these latencies because they seriously deviate from the normal distribution that is required for mean and SE to make any sense. Therefore, the analyses need to be revised, and all mean±SE values for latencies should either be deleted (e.g. Table 1, Fig. 2) or re-calculated as model-predicted values from the revised analyses.
- From Figure 3, it also shows that mean±SE values are inappropriate for the time spent interacting, because there were large outliers. I assume these outliers were the reason that you used the negative binomial distribution for these variables. Unless you calculated the mean±SE values as model-predicted values from the negative binomial analyses, the mean±SE values for interaction times (e.g. Table 2, Fig. 4) should be deleted.
- The total number of problems solved was analyzed with a Poisson model, so again, mean±SE values should either be deleted (L322-323) or calculated as predicted values from the Poisson model.
L348: Is there any reason to expect that any your rats were in "poor metabolic conditions"? This seems very unlikely for those that were maintained in the lab throughout their lives. There could be much more relevant variation among wild-caught individuals, but those differences likely disappeared by the time of testing if they were kept in captivity for at least 7 months before being tested.
L359: What do you mean by "positive state"?
L365-370: Why should lighter be hungrier or more motivated to escape? Body mass by itself does not reveal how under-fed an individual is. Size-corrected body mass indices (like the scaled mass index of Peig & Green 2009-2010) are more relevant in this context. Furthermore, an individual with larger mass may require more food for maintaining homeostasis, so a standardized duration of fasting might be more demanding for them than for light-weight individuals.
L383: Why would captive-born individuals be more motivated to escape?
L385-391: Corticosterone cannot be interpreted as a measure of "metabolism" or "energetic states". These are complex phenomena, and glucocorticoids play a role in regulating them, but there is no simple equation between high or low cort and high or low "metabolism" or "energetic state". For example, high cort levels may increase blood glucose levels (which can be considered an improvement to "energetic state") but at a cost of depleting the glucose stores, leading to a worse "energetic state". Also, it is unclear what you mean by "positive stimulation to environmental conditions".
References:
Gormally, B. M., & Romero, L. M. (2020). What are you actually measuring? A review of techniques that integrate the stress response on distinct time‐scales. Functional Ecology, 34(10), 2030-2044.
Griffin & Guez 2014. Innovation and problem solving: A review of common mechanisms. Behav. Processes 109:121-134.
Peig, J., & Green, A. J. (2009). New perspectives for estimating body condition from mass/length data: the scaled mass index as an alternative method. Oikos, 118(12), 1883-1891.
Peig, J., & Green, A. J. (2010). The paradigm of body condition: a critical reappraisal of current methods based on mass and length. Functional Ecology, 24(6), 1323-1332.
Touma, C., & Palme, R. (2005). Measuring fecal glucocorticoid metabolites in mammals and birds: the importance of validation. Annals of the New York Academy of Sciences, 1046(1), 54-74.
Author Response
Thank you for your feedback. We have addressed the points specifically in the attached document.

Reviewer 2 Report
The authors have made major changes to the background and added missing sections and clarified some things in the materials and methods. The change from stress-related effects to metabolism has made a significant positive impact on the manuscript and the story comes together much better. The authors should, however, still be careful with equating corticosteroid measurements directly with metabolism as they rightly state in the discussion, corticosteroid concentrations could also be affected by other things. As such, high corticosteroid concentrations are not necessarily equal to high metabolism and other measures in addition to glucocorticoids would be needed to determine the true impacts of metabolism. As such, please be careful with statements such as found on l. 102.
Although the manuscript is considerably improved, I am still struggling with the materials and methods and especially with the section describing the tests done. It would be helpful to include the names for the tests (as used later, e.g. l. 263) directly when each test is explained. Further, three separate subheadings are used in the results for each subset of tests (home cage, open field and light-dark box) and it would be useful to follow a similar arrangement in the Materials and Methods. There are also too many different names for the same thing. More specific problems with this are, for example:
- L. 165: I am assuming this refers to the Trixie Dog activity Board (not quite clear). Did the mice have to solve one after the other or were they presented with another task later once they solved one? If the former, how could the solving of one problem affect the subsequent problems?
- L. 175: Is the obstruction task the escape motivated problem (light-dark box problem)?
- L. 175-177: I am assuming that the light/dark box and the Trixie Board were done outside the home cage and this procedure was necessary for both. The different aspects for the different tests should be better combined with each other to make these things more clear.
- Lastly, what was used as a treat for successfully solving the foraging related tasks?
I found the description of the statistics confusing. Please be more precise with which variables were dependent, co-variates and fixed factors. It would probably be helpful to also follow the order provided in the results here. I also don’t quite understand how general linear models were used with negative binomial distributions (parametric tests, which require normal distribution), why these distributions were used for continuous data and why that data was transformed. Also, specifically on lines 253-254, why were different transformations used? In all these cases, the dependent variable is continuous (time), or? The statistics section is still my main concern and this section needs to be reworked and clarifications for the different tests need to be provided.
Now I just have some minor comments, thoughts and corrections that I believe would improve the manuscript further.
- L. 72: Remove “be”.
- L. 155-156: Change to “foraging-motivated problem-solving tests and one escape-motivated problem-solving test.”
- L. 193: Do you have any idea how long the gut passage time is for this species? Knowledge of this would be useful to better understand the impact of possible circadian changes in glucocorticoids on the results. For example, gut passage time would tell the reader if the corticosteroid concentrations measured would be similar to the concentrations when the animals were tested and active. Metabolism would likely be higher during the active phase of the animals and glucocorticoids are also frequently higher during that time and knowledge of this would be important to interpret the results here.
- Table 1: You missed making the changes in the last column for the tile task.
- L. 299: Remove “have”.
- L. 341-343: I am wondering if these two species have very similar ecological characteristics. Striped mice are also highly adaptive occurring in a range of habitats but are also especially successful in semi-desert and desert environments. It may be worth to further investigate and compare these two species.
Author Response

(The authors gave the same response as above.)

Reviewer 3 Report
Questions raised before explanation after modification can be considered and accepted on the premise that supplementary data cannot be provided.
Author Response

(The authors gave the same response as above.)

Reviewer 4 Report
Thanks for revising the manuscript and adding information about the utilized EIA (sorry, but partly to detailed now, because e.g. calculating the concentrations from the optical densities is common practice – please shorten the passage; lines 219-224).
However, my main point was not even addressed (and I doubt it could have been accidentally).
Copy/paste last time: “And it would be mandatory (state-of-the-art) to have this EIA fully validated for the species (FCMs in Melomys cervinipes have not been described so far) under investigation. This needs to include a demonstration that increased (decreased) adrenocortical activity is well reflected in measured FCM concentrations. The latter problem (non-validated method) is even worse, in case of negative findings (like here: no correlation!), because they may be due to the fact that not the right metabolites were measured by the method.” Again, the revised version lacks such a validation of the EIA for the species under investigation (which is now state-of-the-art). I think without this the paper cannot and should not be published.
I’m also not happy with replacing “stress” with “metabolism/metabolic”. Glucocorticoids have metabolic actions, but they are also stress hormones. Maybe better to avoid the problem, by stating that FCMs are a measure of adrenocortical activity?
Below, please find specific comments (ordered by appearance in the ms) – my earlier points are solved now (but the new ones are found in the text added in the revised version)
Line 12: “metabolic processes and hormones” these are not 2 separate factors. Hormones also control metabolic processes.
Line 35: ..using a corticosterone enzyme immunoassay
Lines 37/38: “lower metabolic condition” What is meant here?
Lines 214-215: A similar wording was already present in the previous version: but kits (assays) don’t “extract” the hormone. They may pick it up, or measure it. But adding corticosterone is rather artificial anyway, because it is most likely not present in the faeces. So what does a recovery of 99% tell you?
Lines 217-218: An average CV% between analysed duplicates of 11.4% is very high. Many labs repeat samples when it is >10%. Likewise, an inter-assay CV of 33% is extremely high. So, the measurements were not very precise at all, adding doubts to the methodological quality of FCM measurements.
Line 299: delete “have”
Line 316: as there are only 2 groups, it may be better to give the exact p—value.
Line 328: again, “metabolite” is missing. …faecal corticosterone metabolite concentrations..
Lines 346-350: And most important: the lack of such a relationship could also (likely?!) be attributed to the use of an unsuited method to measure FCMs!
Author Response

(The authors gave the same response as above.)

Round 2
Reviewer 1 Report
Thank you for your continued efforts to revise the manuscript. Several aspects are greatly improved now. I'm happy with the current version of the introduction and discussion, but I'm still concerned about several aspects of the statistical analyses. The changes in the methods have brought up further questions that need addressing.
Detailed comments:
L282-288: For assessing intra- and inter-individual variation, the most relevant analysis here would be a test of repeatability (intra-class correlation), e.g. by the 'rptR' package (see Nakagawa & Schielzeth 2010. Biol. Rev. 85: 935–956.) That would be most informative about the biological significance of your fecal cort measurements collected during cage cleaning (i.e. is it individually consistent?)
L287: The text mentions that the 2020 corticosterone data were log-transformed for the linear model of inter-individual variation. Please clarify in the text if the non-transformed corticosterone data were used in all other analyses that included corticosterone (or if not, what kind of transformation was used). This is an important question because you provide mean±SE values for cort throughout the Results: those values are invalid if the data were non-normal (unless they are back-transformed model estimates).
L291: Please clarify here that two separate analyses were done for each year (this only became clear after reading L340-343). What did you do for individuals that had two measurements? Were they included in both analyses? What are the sample sizes for each year?
L304: Please clarify why you used the 2020 cort data in the analyses of time spent interacting while you used the 2019-2020 cort data for the analyses of solving latencies. Similarly, in L308, please clarify which cort data were used in the analysis of the total number of tasks solved.
L301 and L307: 'lmertest' and 'nlme' are for mixed models, but your Methods text does not mention any random factors. I guess that you analyzed all tasks in a single model, using task type as fixed factor and individual as random factor. Please clarify this in the text. Also, why did you use two different packages when both 'lmertest' and 'nlme' can handle both Gaussian and Poisson data? A side note: the package for 'lmer' is 'lme4' ('lmertest' is an extra, only needed for likelihood-ratio tests and p-value calculations).
L313: Please describe in the Methods whether you checked the model diagnostics. Linear models do not require normality of the data: they require normality of the residuals. They also require homoscedasticity of the residuals, linear relationship with the continuous predictors, and no multi-collinearity. Furthermore, Poisson models require that the dispersion parameter is close to zero. Given that you did not provide access to the data, neither did you present distributional graphs (like boxplots) of the relationships between fecal cort and the studied problem-solving variables, nor did you mention anything about the validity of your statistical models (residual diagnostics), the MS offers very little credibility to your results.
Fig. 1: This graph is redundant because the same values are given in the text (L318-319). Please replace this graph with a boxplot of the original (not transformed) corticosterone values, because that would provide valuable information to the readers and would increase the credibility of your results. I think such a figure would be very important and helpful.
Fig. 2: Again, this graph is redundant because the same values are given in the text (L327). I think this figure can simply be omitted, since the fact that there was a systematic decrease in cort over time does not tell much by itself. This change does not exclude repeatability.
L339: What about the effect of task type?
L340-343: It is nice to see hazard ratios and confidence intervals, but many readers will not understand what those values (eb and […]) are if you do not explain it. And why not show p-values when all other results are presented with p-values?
Fig. 3: I would much rather see a boxplot of the original, non-transformed data than mean±SE values from some hocus-pocus transformation that has no trivial biological interpretation.
Minor phrasing issues:
L212-215: Please clarify this sentence: "Although the gut retention time of mosaic-tailed rats is unknown, collection of faeces in the morning likely corresponded with their main activity period (night time), which is also the period during which problem solving was tested." Are you suggesting here that the feces collected in the morning likely originates from feeding during the preceding night? It is confusing to mention the problem-solving tests here, because it suggests that the fecal samples were collected immediately after the night of a problem-solving test, but that is not true as far as I understand.
L220-225: Normally, these tasks should be described in detail, too. Please add something like "these samples were collected for another study that will be published in a separate paper".
L275-277: "included as dependent variables in the models"
L279: "between the samples collected"
L296-297: "or did not participate (did not interact with the task; cylinder = …)". Also, it needs to be explicitly stated here if the maximal latencies were treated as censored data in the Cox models.
L316: "Biological Validation of Faecal Corticosterone"
L325: To remind the readers that from this on you are talking about "non-stress" fecal samples, please change this sentence to something like: "Corticosterone metabolite concentration measured from faecal samples collected during cage cleaning differed significantly between 2019 and 2020".
Reviewer 4 Report
Please see attached file!

Author Response
Please see attached.

This manuscript is a resubmission of an earlier submission. The following is a list of the peer review reports and author responses from that submission.
Round 1
Reviewer 1 Report
This is an interesting study, but the validity of the presented results cannot be evaluated because there is no mention of the statistical analyses in the Methods. This is a serious shortcoming; I stopped reading the manuscript when I realized this. Looking at Figure 1, it seems that some kind of linear regression was used for the analysis of latencies, which included censored data (maximal latencies); this is inadequate because the latency of solving is not known for individuals that did not solve within the allocated trial time. Such data should be analyzed with methods that can accommodate censored data, such as Cox's proportional hazards models. Statistics that assume a Gaussian distribution, such as mean and SE (as in Table 1) should not be used with non-normal distributions, like latency in Figure 1. Please make sure to use appropriate statistics and provide a detailed, reproducible description of the data analysis. It is very helpful if you make your data (and software code, e.g. R script, if relevant) available if possible.
Further comments:
L15, L20, and throughout: The study was correlational, so all phrasing should be modified accordingly (e.g. change "influence" to "related to" or "correlated with").
L26-27: This contrast between "laboratory animals" and "native animals" is strange. All laboratory animals are (or at least used to be) native somewhere. You studied your animals in the laboratory, and some of them came from the F1-F2 generations of a breeding program (it is not clear from the manuscript whether they were bred in the laboratory). See comments to L112-114, too.
L27-28: "how physiological processes influence behaviour and cognition of native animals is less understood" – this statement is too general. There is an enormous literature on the relationships of physiological processes with behavior in wild animals. Also, throughout the text please pay attention not to equate problem solving (what you studied) with cognition, as these phenomena are not the same and do not necessarily have anything to do with each other; see e.g. Thornton et al. 2014, Papp et al. 2015, Griffin 2016, Griffin & Guez 2014, 2016 (see references at the bottom).
L82-83 and L91: Since you gave a rather narrow definition to problem solving in L77, it is unfortunate that you switch to "innovation" in some sentences. In this context, it is not clear what is meant by "innovation", as this can be a much more general term than the problem solving you define in L77. It may be better to refer to problem solving consistently throughout the text. Alternatively, please define "innovation" and cite justification for using this phrase as synonym for problem solving (e.g. Griffin & Guez 2014).
L93: Throughout the text, it is better to say that you measured problem solving performance or success, rather than ability, because ability may be only one of several aspects that shape performance.
L97-98: These predictions are too simplistic. First, chronic stress can either increase or decrease or have no effect on baseline glucocorticoid levels (Dickens & Romero 2013). Second, the relationship between "stress hormone" levels and cognitive performance is not linear but U-shaped (e.g. Lupien et al. 2009). Failure to recognize these complexities may be a reason for the apparent "equivocal nature of the results from current studies" (L84). I think these complexities should be given careful consideration in the text, especially because you measured corticosterone outside the problem-solving context and it is not known how the test situations changed the hormone levels.
L112-114: Please provide a bit more information to clarify the origin of the study animals. Were the wild-caught individuals captured from a free-living population not participating in the breeding program? How much time did they spend in captivity after capture before starting the study? Is the university's breeding colony maintained in the laboratory or outdoors?
L127-128: What was the duration of fasting before the start of problem solving testing? Was it the same for all individuals?
L135-136: Please provide more information on the timing of tests, in terms of date/season and order within and between individuals. How much time went by between tests per individual, and between individuals per test? Were sexes and birth groups tested in a randomized or systematic order?
L156: What was the starting point from which you measured the latency to solve the puzzle? Did you exclude non-participating individuals (i.e. those who did not interact with the puzzle)?
L165: Please report the number of fecal samples per individual (min-max, mean & SD) and the timing of cleaning sessions relative to the tests. Were the samples collected before/after/during the period of behavioral testing days? Was the timing of fecal sample collection relative to behavior testing standardized across all individuals?
L182: I might be missing something, but where is the description of hormone assay? L171-182 only describes the preparations before the assay.
L185: Please describe how individual mass was calculated for these analyses. How many measurements per individuals? When were these measurements taken relative to the behavioral testing period?
Typos and minor grammar/phrasing errors:
L17: replace "extracted" with "measured"
L48: replace "which" with "and"
L143: puzzle boxes
References:
Dickens & Romero 2013. A consensus endocrine profile for chronically stressed wild animals does not exist. Gen. Comp. Endocrinol. 191:177-189.
Griffin & Guez 2014. Innovation and problem solving: A review of common mechanisms. Behav. Processes 109:121-134.
Griffin & Guez 2016. Bridging the gap between cross-taxon and within-species analyses of behavioral innovations in birds: making sense of discrepant cognition–innovation relationships and the role of motor diversity. Adv. Stud. Behav. 48:1-40.
Griffin 2016. Innovativeness as an emergent property: a new alignment of comparative and experimental research on animal innovation. Phil. Trans. R. Soc. B 371:20150544.
Lupien et al. 2009. Effects of stress throughout the lifespan on the brain, behaviour and cognition. Nat. Rev. Neurosci. 10:434–45.
Papp et al. 2015. A comparison of problem-solving success between urban and rural house sparrows. Behav. Ecol. Sociobiol. 69:471-480.
Thornton et al. 2014. Toward wild psychometrics: linking individual cognitive differences to fitness. Behav. Ecol. 25: 1299-1301.
Reviewer 2 Report
The authors investigated the effects of glucocorticoids, hormones typically elevated during stressful situations, on problem-solving in an Australian rodent, the fawn-footed mosaic-tailed rat. Glucocorticoids can indeed affect learning and problem-solving in animals and depending on the situation, they have positive or negative effects and chronically elevated levels are likely to impair learning and memory. In the present study, however, no effect of baseline glucocorticoids was found on the problem-solving abilities using several different tests. Baseline concentrations are by definition unaffected by stressful situations that could affect problem-solving and it is not clear to me why the authors would expect any effects of baseline concentrations on problem-solving. It is unclear to me if there is maybe a subset of animals with higher and another with lower concentrations where you might expect differences but without providing more information on the measured glucocorticoid concentrations, it is impossible to evaluate this. Alternatively, the authors used wild-caught (but having been in captivity for a while) and captive-bred individuals and indeed found some small differences in problem-solving between them but they do not further evaluate these in regards to glucocorticoid concentrations. Overall, the authors need to provide a better justification for the study and why they would expect any differences with different baseline glucocorticoid levels at all. It is also unclear why this specific species was chosen for this study. What makes this species specifically suitable for this investigation?
The Materials and Methods and Results are missing important information on the glucocorticoid measurements and statistics without which an evaluation of the results and conclusions is impossible.
- More data on the baseline glucocorticoid concentrations should be provided. What was the average concentration and how much did it vary between individuals (and groups)?
- Problem Solving Tests: Please provide a better outline of the different tests done. From the description, it seemed like only four tests were done, but later (l. 154-155) six tests were mentioned and ultimately there were 7. Also, a little more information on each of the tests should be added, e.g test was done in a home cage or the animal was moved to another cage. If the latter, was the animal just dropped in or how was it introduced to the problem?
- L. 155-157: Please provide more information on how these variables were measured. Was a specific program used? Further, from when to when were times measured, i.e. was latency to solve the puzzle measured from the time the animal was exposed to the test (put in a cage) or from when the animal started to interact with it?
- L. 167-170: Because of the gut passage time, sampling during cage cleaning shouldn't have any effect on the faecal results except if cage cleaning took very long. That is the beauty of using faeces, which could probably be better highlighted. How often were cages cleaned (and samples taken) and was it done at similar times or different times of the day?
- Much more information on the faecal glucocorticoid measurements is needed. What assay was used? What were the sensitivity, intra- and inter-assay coefficients of variation, parallelism, etc.?
- Most importantly, faecal analysis for glucocorticoids needs to be validated for the species first. At the moment, it is totally unclear if what was measured in this study was actually glucocorticoid metabolite concentrations associated with stress. Without validation, the results can't be evaluated at all.
- No section on data and statistical analysis is provided. As stats are provided in the results, this still needs to be included.
- L. 238: It is unclear to me why you would assume that your animals were chronically stressed. Moreover, baseline glucocorticoid concentrations were measured in this study and none associated with acute or chronic stress. Why would you assume that higher corticosterone concentrations in your animals are related to chronic stress? It could just be individual variation and considering that faecal hormones were measured, those individual variations could be associated with digestion and metabolism and not show actual differences in hormone concentrations. Also, it is difficult to ascertain effects without an idea of how much concentrations varied between individuals (see above).
- L. 277: It doesn’t make sense to assume that learning and memory played a role as only problem solving but not learning and memory was evaluated.
Reviewer 3 Report
All in all, this is a meaningful study. Explain the relationship between fecal hormone levels and behavior. This paper has clear logic and reasonable statistical methods, but for animal behavior research, 24 samples are too small, so the R2 and P values of most results, especially the regression equation, are not reliable. Therefore, the conclusion drawn by the author is only the result of a small sample and a small range. Is it representative?
Reviewer 4 Report
The paper of Rowell et al submitted to “Animals” evaluates the relation between fecal corticosterone metabolites (FCMs) and five different problem solving tasks in the fawn-footed mosaic-tailed rat, but found no correlations.
I have some serious concerns about the paper. First, the aim (reflected in the title) is set wrong: Why should FCMs have an influence on problem solving at all? FCMs are the excretory product of plasma corticosterone, and they are used as a non-invasive measure of adrenocortical activity, and plasma corticosterone levels. Thus, they cannot influence anything, maybe plasma corticosterone can influence, but even here the relation between glucocorticoids (GC) and behavior is a very complex one (GC may influence behavior and vice versa, but in addition, both may be influenced for example by other processes within the central nervous system during a stress response). So it is a problem of cause and effect, which is not easily resolved. In addition, the method for measuring FCMs is not fully described. It stops after the extraction (have all authors seen the submitted version?). However, it is crucial which enzyme immunoassay was utilized. And it would be mandatory (state-of-the-art) to have this EIA fully validated for the species (FCMs in Melomys cervinipes have not been described so far) under investigation. This needs to include a demonstration that increased (decreased) adrenocortical activity is well reflected in measured FCM concentrations. The latter problem (non-validated method) is even worse, in case of negative findings (like here: no correlation!), because they may be only due to the fact that not the right metabolites were measured by the method. I also do not think that conclusions drawn in this study are based on results (likely, it’s due to a simplistic equation: higher GC = stress).
Thus, in its present form (and without the missing points) the manuscript is not suited for publication.
Below, please find specific comments and questions (ordered by appearance in the ms):
Line 12: Replace “corticosteroids” with “glucocorticoids” (the former would also include aldosterone).
Line 15: It’s not about strains, but species.
Line 17 sounds strange and in this sentence several things are wrong (e.g.: EIAs are not used to extract hormone concentrations).
Line 21-23: How do you come up with the conclusion that rats were resilient to stress?
Line 24/25: The sentence sounds as if the changing GC are the challenge. Besides, GCs don’t change, but only their levels.
Line 26 (and 28): Same problem as outlined above: FCM concentrations don’t influence something.
Line 34: …using a corticosterone enzyme immunoassay.
Line 97/98: What is the basis of this assumption? May you not argue to the opposite? In the literature of different coping styles (reactive vs proactive), reactive individuals tend to have higher baseline (and stress response) GC levels, but they are those which higher problem solving abilities (e.g. see Stöwe et al., 2010 for personality traits and FCMs).
Line 98-99: And a situation of high stress is something different, you measured baseline levels.
Line 158 onwards: Which time of day were samples collected (the same for all individuals?)
Lines 167-170: I cannot follow the argument here. FCM levels reflect the situation a certain time before (several hours, but depending upon the species – one purpose of a sound validation of the method to measure FCMs is also to evaluate this delay times in the investigated species). Therefore, FCM levels in samples collected during the behavioural test would also not being influenced by the test itself.
Line 182: “This dilution was used as the sample for hormone extraction.” This is a very strange sentence making absolutely no sense!
A paragraph about the utilized method (corticosterone EIA) to measure FCMs is totally missing, but crucial to evaluate the whole paper properly. In addition, details about applied statistics are also missing? Which tests (and software) were used? Did you correct for multiple measurements?
Line 187; 189 (203 and elsewhere): As outlined above: FCM levels cannot impact/influence something
Line 192 (Fig 1): Ticks of the axes are missing. It seems as if there are two sub-groups with different weights – are those females/males, which most likely have different weights, or? And the dots at a line at 1800 sec? They resemble animals which probably did not solve the task until the end of the experiment - 30 min, or?
Lines 241 onwards: Ref 41 is exactly about personality traits and GC. The approach in this ms that higher levels of corticosterone equals chronic stress is simple wrong. The picture is much more complicated (e.g. see MacDougall-Shackleton et al., 2019)
References cited above:
MacDougall-Shackleton, SA., Bonier, F., Romero, LM., Moore, IT. (2019): Glucocorticoids and “stress” are not synonymous. Integr. Organismal Biol. 1, obz017. https://doi.org/10.1093/iob/obz017
Stöwe, M., Rosivall, B., Drent, P., Möstl, E. (2010): Selection for fast and slow exploration affects baseline and stress induced corticosterone excretion in Great tit nestlings, Parus major. Horm. Beh. 58, 864-871. https://doi.org/10.1016/j.yhbeh.2010.08.011